# Cholesterol Levels Affect the Performance of AuNPs-Decorated Thermo-Sensitive Liposomes as Nanocarriers for Controlled Doxorubicin Delivery

**DOI:** 10.3390/pharmaceutics13070973

**Published:** 2021-06-27

**Authors:** Mónica C. García, Nabila Naitlho, José Manuel Calderón-Montaño, Estrella Drago, Manuela Rueda, Marcela Longhi, Antonio M. Rabasco, Miguel López-Lázaro, Francisco Prieto-Dapena, María Luisa González-Rodríguez

**Affiliations:** 1Departamento de Ciencias Farmacéuticas, Facultad de Ciencias Químicas, Universidad Nacional de Córdoba, Ciudad Universitaria, Haya de la Torre and Medina Allende, Science Building 2, Córdoba X5000HUA, Argentina; mrlonghi@unc.edu.ar; 2Unidad de Investigación y Desarrollo en Tecnología Farmacéutica, CONICET, Consejo Nacional de Investigaciones Científicas y Técnicas, UNITEFA, Córdoba X5000HUA, Argentina; 3Department of Pharmacy and Pharmaceutical Technology, Faculty of Pharmacy, Universidad de Sevilla, C/Prof. García González 2, 41012 Seville, Spain; poichicha86@hotmail.fr (N.N.); amra@us.es (A.M.R.); 4Department of Pharmacology, Faculty of Pharmacy, Universidad de Sevilla, C/Prof. García González 2, 41012 Seville, Spain; jcalderon@us.es (J.M.C.-M.); mlopezlazaro@us.es (M.L.-L.); 5Department of Physical Chemistry, Faculty of Chemistry, Universidad de Sevilla, C/Prof. García González s/n, 41012 Seville, Spain; star_9_trebol@hotmail.com (E.D.); marueda@us.es (M.R.); dapena@us.es (F.P.-D.)

**Keywords:** gold nanoparticles, temperature-sensitive nanocarriers, anchoring, controlled drug release, Langmuir monolayers, liposomal formulations, anticancer activity

## Abstract

Stimulus-responsive liposomes (L) for triggering drug release to the target site are particularly useful in cancer therapy. This research was focused on the evaluation of the effects of cholesterol levels in the performance of gold nanoparticles (AuNPs)-functionalized L for controlled doxorubicin (D) delivery. Their interfacial and morphological properties, drug release behavior against temperature changes and cytotoxic activity against breast and ovarian cancer cells were studied. Langmuir isotherms were performed to identify the most stable combination of lipid components. Two mole fractions of cholesterol (3.35 mol% and 40 mol%, L1 and L2 series, respectively) were evaluated. Thin-film hydration and transmembrane pH-gradient methods were used for preparing the L and for D loading, respectively. The cationic surface of L allowed the anchoring of negatively charged AuNPs by electrostatic interactions, even inducing a shift in the zeta potential of the L2 series. L exhibited nanometric sizes and spherical shape. The higher the proportion of cholesterol, the higher the drug loading. D was released in a controlled manner by diffusion-controlled mechanisms, and the proportions of cholesterol and temperature of release media influenced its release profiles. D-encapsulated L preserved its antiproliferative activity against cancer cells. The developed liposomal formulations exhibit promising properties for cancer treatment and potential for hyperthermia therapy.

## 1. Introduction

Advanced nanomaterial therapeutic systems have several advantages over conventional therapeutic systems. Targeted delivery of drugs to specific sites of action constitutes a promising alternative in order to overcome the current limitations of drugs [1] and comprises a big challenge for researchers in the field of pharmaceutical sciences, which have witnessed great progress in the development of improved drug delivery systems, including lipid-based and polymer-based carriers [2,3].

Among lipid-based carriers, liposomes (L), since their first use as drug delivery systems in the 1970s [4], have risen as one of the most useful tools for drug delivery in the field of nanomedicine [5,6]. They offer several advantages including biocompatibility, capacity for self-assembly, ability to carry large drug payloads, increased half-life, low toxicity, good solubilization and stability of encapsulated drugs, and preventing the degradation of the payload in the physiological environment, as well as a wide range of physicochemical and biophysical properties that can be modified to control their biological performance [5,7]. However, L have demonstrated several pharmacological problems; among them, a major one is related to the ability of the mononuclear phagocyte system in the host immune system to recognize them, and subsequently rapidly clear them by the reticuloendothelial system in the liver and spleen. Therefore, in order to overcome these shortcomings, novel strategies have been explored to design liposomal systems that respond to a specific stimulus to release their payload in the target site [7], which is particularly useful in cancer therapy.

Tumor tissues usually exhibit hyperthermia because of their rapid metabolism, and fever frequently occurs in the tumor sites. This phenomenon has inspired the development of temperature-sensitive L as nanocarriers to improve cancer pharmacotherapy [8]. Their responsiveness allows for control of drug release, which is a key contributor to the efficacy of nanomedicines [9]. In the case of temperature increasing, this type of system can release the anticancer payload at the tumor site under the condition of pathological hyperthermia or external warming by which solid tumors can also be heated by a controlled device with an external energy source [10]. Temperature-sensitive L may improve tumor accumulation, trigger liposomal drug bioavailability, enhance drug delivery and drug internalization. They also minimize the toxicity of the encapsulated anticancer agent, control the release rate and enhance the long circulating properties by modulating their composition and their behavior as smart nanocarriers [11].

Temperature-responsive L can be prepared with phospholipids that show a gel-to-liquid crystalline phase transition a few degrees above physiological temperature [12]. Among them, 1,2-dipalmitoyl-*sn*-glycero-3-phosphocholine (DPPC) has been widely used for obtaining these temperature-sensitive liposomal systems due to its low phase transition temperature, which is around 41 °C [1]. Moreover, hybrid nanomaterials composed of both inorganic/organic components have recently been studied as promising platforms for therapeutic applications, which also have been explored in the development of temperature-responsive hybrid nanocarriers. They combine the mechanical and thermal stability of the inorganic component and the ease of processability of the organic substance [13,14]. One strategy that has been investigated to develop hybrid nanosystems is the use of gold nanoparticles (AuNPs) combined with L, which were firstly postulated by Paasonen and cols. [15]. AuNPs exhibit unique optical/physical attributes and have tunable optical properties because of their localized surface plasmon resonance when exposed to infrared electromagnetic radiation, transforming most of the absorbed energy to heat [16].

Among the different approaches reported, the anchoring of AuNPs onto the surface of L provides several advantages, including detection and imaging. Also, AuNPs-functionalized L can be used to activate the drug release in virtue of local temperature changes that the inorganic component can experience as a light-induced heating response, which could positively impact in the intracellular delivery and cell damage in cancer disease [9,17,18,19]. In this regard, light illumination is considered a useful tool since it may induce chemo-physical changes in externally triggerable nanomedicines, improving their biological performance [9]. To anchor AuNPs onto the surface of L, an appropriate modification of bilayer composition to confer them surface charge needs to be performed. Lipids containing opposite charges have been considered as a component of L [1] and didodecyldimethylammonium bromide (DDAB), a synthetic cationic lipid, has been employed. It is a homolog double-chain useful to prepare lipid vesicles in aqueous media and other membrane models, and it has been widely studied since it can form stabilized cationic L [1,20,21]. Moreover, the presence of cholesterol plays a key role when stabilized L are prepared. It is well-known that cholesterol has effects on the structural and dynamic properties of synthetic and natural membranes [22]. The incorporation of cholesterol contributes to bilayer rearrangement. It modulates several properties of L, namely rigidity and fluidity of membranes [23,24], thickness [25] and stability [26]. Furthermore, its proportion with respect to the phospholipid in membranes affects the drug encapsulation efficiency [27,28,29]. However, there are few reports regarding the ratio between cholesterol and phospholipid used in L formulation [26,28,30,31,32], and there are even fewer regarding the ratio cholesterol/phospholipid to provide a controlled drug release [26,28,30].

We hypothesized that cholesterol levels play a key role in the development of cationic L composed of DPPC and DDAB, anchored with negatively charged AuNPs and that its levels would impact the physicochemical and pharmaceutical properties of AuNPs-decorated drug-encapsulated temperature-responsive L. Therefore, we consider that it would be valuable to define the best combination of DPPC and cholesterol that allows obtaining thermodynamically stable L with the required controlled release of the encapsulated molecule.

In this study, doxorubicin (D), an anthracycline antibiotic originally isolated from *Streptomyces peucetius* var. *caesius*, was selected since it is one of the most commonly employed anticancer drugs. This drug is widely used in the treatment of solid and hematologic neoplasms, such as breast, ovarian, bladder cancer and lymphoma [33]. It causes DNA damage by intercalating between the DNA base pairs, blocking DNA-topoisomerase II, and generating free radicals. All these effects, and other mechanisms involved in its cytotoxicity, allow it to kill rapidly growing cells, such as cancer cells. However, this antineoplastic molecule causes many adverse events, highlighting its high cardiotoxicity [34,35,36]. Therefore, optimized D-containing formulations are required to improve cancer therapy.

The goal of the present work was to evaluate the effects of cholesterol levels in the performance of AuNPs-functionalized cationic and thermo-sensitive L composed by DPPC and DDAB as nanocarriers for controlled D delivery as well as their drug release behavior against temperature changes. Their cytotoxic activity against breast (MDA-MB-231) and ovarian (SK-OV-3) cancer cells was studied. Anionic AuNPs were selected to functionalize L since they are less cytotoxic than the cationic ones [37]. Cationic L were prepared since they can improve the electrostatic interaction with the anionic AuNPs [1,38].

## 2. Materials and Methods

### 2.1. Materials

1,2-Dipalmitoyl-*sn*-glycero-3-phosphocholine (DPPC) was supplied by Avanti Polar Lipids (Alabaster, AL, USA), and cholesterol and Sephadex^®^ G-50 were obtained from Sigma Aldrich^®^ (Barcelona, Spain). Gold nanoparticles (AuNPs, 15 nm) solution was obtained from Nanovex Biotechnologies^®^ (Asturias, Spain). Didodecyldimethylammonium bromide (DDAB) was provided by Sigma-Aldrich^®^ (Barcelona, Spain). Sephadex G-50 beads were provided by Sigma Aldrich^®^ (Barcelona, Spain).

The reagents NaCl, NaOH pellets, Na_3_PO_4_, KH_2_PO_4_, K_2_HPO_4_ and ammonia 30% (PA grade, PanReac AppliChem^®^, Barcelona, Spain), Hepes for buffer solutions (PA grade, PanReac AppliChem^®^, Barcelona, Spain), chloroform stabilized with amylene (EPR grade, LAbKem, Barcelona, Spain), doxorubicin (D) and Triton X-100 solution (Sigma Aldrich^®^, Barcelona, Spain), HCl 37% (PA grade, T3 Química, Barcelona, Spain), (NH_4_)_2_SO_4_ and NH_4_H_2_PO_4_ (Sigma Aldrich^®^, Barcelona, Spain) and C_6_H_14_N_2_O_7_ (PA grade, Fluka^®^, Barcelona, Spain) were used as purchased without further purification. Phosphate buffer pH 7.4 (to simulate plasmatic conditions) was prepared according to the United States Pharmacopeia specifications [39] using analytical grade reagents.

All experiments were carried out with distilled and purified water.

### 2.2. Cell Cultures

SK-OV-3 cells (human ovarian cancer cells) were purchased from the Cell Line Service (CLS, Eppelheim, Germany). MDA-MB-231 cells (human breast cancer cells) were purchased from the American Type Culture Collection (ATCC). Cells were maintained in Dulbecco’s Modified Eagle’s Medium (DMEM) supplemented with 10% fetal bovine serum, 100 U/mL penicillin and 100 μg/mL streptomycin. All cells were kept in a humidified 37 °C, 5% CO_2_ incubator. All cell culture reagents were purchased from Biowest (Nuaillé, France).

### 2.3. Solubility Studies of Doxorubicin

These studies were carried out by MultiScreen^®^ solubility assay. Aqueous solutions of ammonium salts of sulfate, hydrogen phosphate and citrate, at pH 4 and 7, were prepared. The pH values were adjusted with ammonia or hydrochloric acid. Four hundred microliters of the salt solutions (300 mM) were added to 100 µL of D aqueous solution (3.45 mM). The samples were mixed on a plate shaker at 50 rpm over 2 h at 25 °C (Bath 320 OR Unitronic OR, Selecta P^®^, Barcelone, Spain), and then were filtered at vacuum in a filtration equipment (Millipore, MultiScreen^®^ HTS Manifold, MA, USA).

#### 2.3.1. Doxorubicin Quantification by HPLC-Fluorescence Detection

D content was determined by high-performance liquid chromatography (HPLC) analysis using fluorescence detection. The analytical quantification of D in the samples was performed using a Hitachi^®^ LaChrom Elite HPLC system (Hitachi High Technologies America, Inc., San Jose, CA, USA)equipped with an isocratic pump and an L-2485 fluorescence detector at 475 and 580 nm of excitation and emission wavelengths, respectively, with data acquisition and processing being performed using a EZChromElite Data System Manager software (3.1.3. version, Hitachi High Technologies America, Inc., San Jose, CA, USA). Chromatographic separations were carried out using a Zorbax SB^®^ C18 reverse phase column (4.6 × 150 mm, 3.5 µm particle size, Agilent, CA, USA). Analysis was performed with methanol and 0.1% of ammonia solution adjusted to pH 3.0 with formic acid (70:30, *v*/*v*) as the mobile phase at a flow rate of 1 mL/min in the isocratic mode. All solutions were filtered through 0.45 µm cellulose filters (Millipore^®^, Barcelone, Spain) and then quantified by HPLC. For analysis, 20 µL of each sample were injected, and the run time was set at 6 min, with the retention time being observed at 3.5 min. D concentrations in the samples were quantified by relating the analytical peak area to the regression line of the calibration curve. For the calibration curve, a stock solution of 10 mg/mL D in distilled water was prepared. Subsequently, a six-point calibration curve was prepared over a range of 1.2 to 6.0 mg/mL by diluting several milliliters of stock solution with mobile phase. These solutions were prepared in triplicate. The calibration curve was constructed by plotting the mean peak area of D vs. D concentration.

#### 2.3.2. Morphological Characterization of Doxorubicin Crystals by SEM

Scanning electron microscopy (SEM) analyses (Philips^®^ XL 30, Philips, Eindoven, The Netherland) were carried out to D crystals after solubility studies. Briefly, before the examination, the samples were coated with sputtered Au-Pd in Ar atmosphere in a high vacuum evaporator. Images were then obtained at an excitation voltage of 20 kV.

### 2.4. Preparation of Monolayers at the Air/Water Interphase and Measurement of the Langmuir Isotherms

A Nima Langmuir trough, model 611D, made in PTFE (270 cm^2^ of area and c.a. 150 cm^3^ of subphase volume) was used for measuring the Langmuir isotherms. The trough was equipped with two PTFE moving barriers and a PS4 Nima pressure sensor. The Wilhelmy plate was a strip of 1 cm-wide Whatman^®^ chromatographic paper renewed in every measurement. The trough was cleaned with methanol and rinsed with Milli Q water several times before the preparation of every monolayer. In order to avoid contamination from the laboratory atmosphere, the trough was protected in a perspex cabinet. The isotherms consisted of the representation of the surface pressure (π) versus the averaged area per lipid molecule at the monolayer (A_molec_). The surface pressure is the difference between the surface tension at the air/water interface in the absence and in the presence of the monolayer: π = γ_0_ − γ.

Stock lipidic solutions (of DPPC, cholesterol and DDAB) were prepared in chloroform with an approximate concentration of 1 mg/mL. The molar ratios desired for every monolayer were prepared by mixing the calculated volumes of each stock solution. All the glassware used for the lipidic solutions was cleaned in piranha solution overnight, carefully rinsed with milli-Q water and dried in an oven at 80 °C.

Once the trough was cleaned, the trough top was filled with freshly collected milli-Q water with the PTFE barriers in an opened position. The barrier position was set to the closed one, and the surface of the subphase between them was vacuum-aspirated with a pipette tip to remove any dust on the surface. Then, the barriers were again opened. A volume (20–30 µL) of the lipid mixture solution with the desired mole fraction in chloroform was spread over the interphase with a Hamilton syringe. The solvent of the lipid solution was evaporated for 20 min. Then, the barriers compressed the monolayer at a rate of 25 cm^2^/min, while the isotherms were registered at lab temperature (22 ± 1 °C).

### 2.5. Liposome Preparation

Pre-weighed components, DPPC, cholesterol and DDAB, at two different ratios—75.24:3.35:21.42 mol% and 45:40:15 mol%—to prepare two different series of L, namely L1 and L2, respectively, were dissolved in chloroform. The samples were evaporated (Büchi^®^, R-200) at 42 °C to remove the organic solvent until obtaining a thin lipid film, which was then hydrated by adding 3 mL of 250 mM (NH_4_)_2_SO_4_ aqueous solution. Multilamellar L were formed after five alternated cycles consisting of stirring by vortex for 1 min and heating bath at 42 °C for 5 min.

Two milliliters of liposomal dispersions were then placed in an extruder (Avanti Polar Lipids^®^, Alabaster, AL, USA) under airflow, adjusted to 58 °C in a heating bath and extruded through 0.8 µm polycarbonate membranes followed by 0.2 µm polycarbonate membranes 3 times each [40] to obtain unilamellar L.

A transmembrane ammonium sulfate gradient between vesicle core and external media or continuous phase was achieved by the dialysis method [41,42]. For that, the obtained unilamellar liposomal dispersions were taken inside a cellulose dialysis bag (10 kDa) and dialyzed against Hepes buffer solution pH 7.4 at a volume ratio of 1:600 for 6 h, at room temperature (23–25 °C), and under magnetic stirring.

Appendix A shows a schematic depiction of L preparation.

### 2.6. Doxorubicin Encapsulation

D was encapsulated into L by remote loading method [43]. First, L were incubated in a heating bath at 42 °C for 5 min. Pre-determined amounts of liposomal sample and D aqueous solution at a concentration of 1 mg/mL were mixed by adding small aliquots of D to liposomal dispersions, vortexing for 1 min and incubating at 42 °C. After adding D, the samples were incubated for 1 h in a heating bath at 42 °C. Then, the samples were left overnight at 4–8 °C for further purification.

The unloaded D was removed by Sephadex^®^ G50 column. For that purpose, Sephadex^®^ G-50 beads were hydrated in previously degassed Hepes buffer at pH 7.4. Two-milliliter syringes without needles were used to prepare the columns. A small portion of glass wool was rolled into a small ball and firmly inserted into an empty syringe using the plastic plunger, thus creating a plug for the column. Each column was placed into an empty glass tube and filled with the hydrated Sephadex^®^. Columns were gently shaken to compact the Sephadex^®^ G-50 beads, removing air bubbles, and more Sephadex^®^ beads were added to fill each column with 2 mL of beads. The excess of Hepes buffer was removed from the glass tube by gravity. Columns were then placed into clean test tubes and were ready to be used within 1 h of preparation. Before adding the liposomal dispersions, the columns were rehydrated with 200 µL of Hepes buffer, and 300 µL of the sample was immediately added to each column. Aliquots of 200 µL of Hepes buffer were added to keep a constant flow of eluted fluid. The D-loaded L were eluted in the first 1.5 mL and were collected for further studies. Free D was collected after adding 3 mL of Hepes buffer solution for elution.

Appendix A shows a schematic depiction of D encapsulation into L.

### 2.7. Functionalization with AuNPs

AuNPs surface anchored-liposomes (AuNPs-L) and doxorubicin-loaded AuNPs surface anchored-liposomes (AuNPs-L-D) complexes were prepared by adding an AuNPs suspension of 15 nm diameter to the liposomal dispersion at 8:3 (*v*/*v*) ratio and stirred with a vortex for 1 min.

Appendix A shows a schematic depiction of the anchoring of AuNPs once obtained the D-loaded L.

### 2.8. Characterization of Liposomal Formulations

#### 2.8.1. Size and Surface Charge

The hydrodynamic apparent diameter (d_H_) and zeta potential (ζ) of the samples were determined by photon correlation spectroscopy (dynamic light scattering, DLS) and electrophoretic light scattering (ELS) measurements, respectively, using a Zetasizer Nano-S instrument (Malvern Instruments^®^, Malvern, UK). The d_H_ values were calculated by the method of cumulants, and size distributions were obtained by the CONTIN algorithm. Polydispersity indexes (PDI) of liposomal dispersion were also determined. Electrophoretic mobilities were converted to ζ using the Smoluchowski equation. All measurements were performed in triplicate at 25 °C with the samples dispersed in Hepes buffer.

#### 2.8.2. Morphological Analysis

Field-emission scanning electron microscopy (FE-SEM) analyses (Hitachi^®^ S5200, Hitachi, Krefeld, Germany) were carried out to the obtained liposomal formulations. Briefly, before morphological evaluation, 40 µL of dispersion was placed in a Si sample holder and observed (uncovered samples). Images were then obtained at an excitation voltage of 5 kV. Samples coated with sputtered Pt in Ar atmosphere in a high vacuum evaporator were also analyzed (covered samples).

Liposomal formulations were also examined by transmission electron microscopy (TEM, Libra 120, Carl Zeiss AG, Oberkochen, Germany). A drop of each sample was placed on a formvar/carbon-supported copper grids, and the material excess was removed with a filter paper. A 1% *w*/*v* uranyl acetate solution was dropped onto the grid, used as a negative staining agent. The excess staining solution was removed after washing twice, and the grid was left to dry. Finally, the prepared grid was examined under the TEM microscope at 120 kV.

#### 2.8.3. UV-Visible Spectroscopy Studies

Absorption UV-Vis spectra of the liposomal formulations and AuNPs suspension were recorded with a spectrophotometer Agilent^®^ 8453, from 400 to 800 nm, with 0.5 nm of resolution. Spectra are shown in Appendix A.

#### 2.8.4. Encapsulation Efficiency

The amount of encapsulated D in the liposomal dispersions was determined by absorbance measurements at 480 nm (Agilent^®^ 8453 System, Agilent, Victoria, Australia) after lysis of L with Triton X-100 (final concentration 0.5% *v*/*v*). The encapsulation efficiency (EE %) was determined according to Equation (1):(1)$$\mathrm{EE}\;\%=\frac{D_{l}}{D_{t}}\times 100$$
where D_t_ is the total amount of drug added to prepare the D-containing L and D_l_ is the amount of doxorubicin quantified after lysis of L.

The EE % was also indirectly determined by quantifying D solution samples eluted in the purification process.

### 2.9. In Vitro Release Studies

D-containing liposomal dispersions were subjected to drug release analysis. The release rate from an aqueous solution with an equivalent concentration of D was used as a reference. Experiments were performed through a dialysis method. In these studies, 1 mL of each liposomal dispersion was placed in the dialysis bag and both borders were sealed with a dialysis clip. The dialysis bag (molecular cut-off of 10 kD) was incubated in 50 mL of release medium (Hepes pH 7.4). The whole setup was placed in an automated shaker, protected from light and gently stirred (80 rpm) at 37 °C and 42 °C. Samples of 0.5 mL of receptor medium were withdrawn at predetermined time intervals and replaced with equal quantities of fresh medium. The concentration of released D was assayed in a fluorescence spectrophotometer (Hitachi^®^ F-2500, Hitachi High Technologies America, Inc., Pleasanton, CA, USA) at 480 and 580 nm of excitation and emission wavelengths, respectively, and analyzed in an FL Solutions V.1. software. All experiments were carried out in triplicate and sink conditions were maintained. The cumulative percentage of D released was calculated and expressed as a function of time. The results were expressed as the % average of three determinations with their SD.

The mean release profiles were fitted according to common mathematical models as follows [44,45]:(2)$$\frac{D_{t}}{D_{0}}=k_{Z}\times t$$
(3)$$\frac{D_{t}}{D_{0}}=k_{H}\times t^{0.5}$$
(4)$$\frac{D_{t}}{D_{0}}=k_{P}\times t^{n}$$
where *D*_t_ (%) is the percentage of drug released at time t; *D*_0_ is the initial value of *D*_t_, t is the time; *n* is the diffusion release exponent; and *k_Z_*, *k_H_*, and *k_P_* are the kinetic constants corresponding to Zero order, Higuchi and Korsmeyer–Peppas kinetic models, respectively.

### 2.10. Cell Viability Assay

Cell viability was evaluated by the resazurin assay. This assay is a colorimetric technique based on the reduction of blue compound resazurin by viable cells into the pink soluble product resorufin. The quantity of resorufin produced is proportional to the number of viable cells. Briefly, exponentially growing cells were seeded in 96-well plates and allowed to grow for 24 h. The cells were then exposed to the treatments for 2 h and were allowed to grow for an additional 70 h in drug-free medium or for 72 h. The treatments evaluated were free D, L2-D and AuNPs-L2-D in a drug concentration range of 0–600 ng/mL. After the treatment period, cells were washed once with PBS, and 150 μL resazurin (20 µg/mL in medium) were added to each well. Plates were incubated for 6 h at 37 °C and 5% CO_2_, and, finally, optical densities were measured at 540 nm and 620 nm with an absorbance spectrophotometer microplate reader. Cell viability was expressed as a percentage respect to the untreated cells. The results were expressed as the mean ± standard error of the mean (SEM). All data are from two independent experiments.

## 3. Results

To prepare temperature-responsive L and to evaluate the effect of cholesterol levels on L and AuNPs-functionalized L, DPPC, DDAB and two different proportions of cholesterol were used to prepare lipid-based nanocarriers, which were further functionalized with AuNPs and loaded with D, selected as anticancer drug model. Eight different samples were obtained, named as described in Table 1, depending on their compositions.

### 3.1. Solubility Studies

To select the salt solution for D loading, solubility studies of D in different ammonium salts (citrate, phosphate and sulfate) at two different pH values (4 and 7) were evaluated. This solubility study aimed to select the less soluble ammonium salt-D for achieving a prolonged drug release when loaded into L. As observed in Figure 1, solubility results showed that in ammonium sulfate solutions, the solubility of D was independent of the pH evaluated (4 and 7), whereas, in phosphate and citrate buffer, D solubility was pH-dependent. In these cases, the solubility of D decreased at high pH values. Therefore, at pH 4, no D precipitates were formed, while at pH 7, the solubility was lower than at pH 4 and important precipitates were formed, corresponding to D crystals.

SEM images revealed the different morphological characteristics of D precipitates depending on the salt solutions evaluated and their influence of pH on particle size and morphology (Figure 2). As observed, in sulfate at pH 4 (Figure 2A), the precipitates were irregular; some of them were elongated with a smooth surface, and others were smaller in size. However, in this salt solution and at pH 7 (Figure 2B), the morphology was different; larger particles with irregular shape and non-smooth surfaces were observed. In citrate salt at pH 7 (Figure 2C), the precipitates were smaller and with lower aggregation than those obtained in sulfate at the same pH value. In phosphate salt at pH 7 (Figure 2D), the precipitates were larger than in citrate salt, with small particles adhering to their surface.

### 3.2. Langmuir Isotherms of DPPC:DDAB:Cholesterol Ternary Monolayers

Figure 3A contains the Langmuir isotherms corresponding to monolayers of pure DPPC, DPPC:DDAB in a 3:1 molar ratio, pure cholesterol and ternary monolayers of DPPC:DDAB 3:1 and cholesterol with selected mole fractions of cholesterol. The monolayer corresponding to the pure DPPC showed the liquid expanded (LE) to liquid compressed or condensed (LC) plateau at a surface pressure (π) of c.a. 10 mN/m. The 3:1 molar ratio in the mixed monolayer DPPC:DDAB was selected to provide the lipidic film with the positive charge enough for the efficient anchoring of negatively charged AuNPs. The presence of DDAB extended the LE region of the isotherm to higher surface pressures and originated a decrease in the compression slope. On the contrary, the isotherm corresponding to a monolayer of pure cholesterol did not show the LE region, and only a compressed phase could be observed. The presence of cholesterol in the ternary monolayers eliminated the LE-LC transition in the isotherms and increased their slope.

The effect of cholesterol in the elastic properties and phase transitions of the monolayer can be more clearly seen in the elastic compression modulus (1/*C_s_*) defined as:(5)$$\frac{1}{C_{s}}=-A_{\mathrm{molec}}\left(\frac{d\pi }{dA_{\mathrm{molec}}}\right)$$
where *C_s_* is the elasticity modulus and A_molec_ is the area per molecule in the monolayer.

Figure 3B shows the dependence of the elastic compression modulus with the surface pressure corresponding to the isotherms in Figure 3A.

The plots in Figure 3B show that the inclusion of DDAB initiated a decrease in the maximum 1/*C_s_* of the DDPC monolayer, from 175 mN/m to c.a. 110 mN/m. The presence of cholesterol at mole fractions higher than 0.3 increased the 1/*C_s_* up to values close to 600 mN/m.

In order to select the optimum lipidic formulation for the L, it is convenient to analyze the thermodynamics of the ternary monolayer, which could provide information about the stability of the lipidic films involved in both supramolecular structures. In Figure 4A, the area per molecule was plotted vs. the mole fraction of cholesterol at three surface pressures. In the case that the molecules of cholesterol mix ideally with the other components of the monolayer or if cholesterol forms independent domains in the monolayer, the area per molecule obtained would obey Equation (6):(6)$$A_{\mathrm{molec}}^{\mathrm{ideal}}=x_{\mathrm{chol}}A_{\mathrm{molec}}{}_{{(x}_{\mathrm{chol}}=1)}{+(1-x}_{\mathrm{chol}}{)A}_{\mathrm{molec}}{}_{{(x}_{\mathrm{chol}}=0)}$$

Then, the resulting plot will be a straight line connecting the two extremes of the plot, at x_chol_ = 0 and x_chol_ = 1.

The 1/C_s_ vs. x_chol_ plotted at constant surface pressure in Figure 4B show positive deviations from the ideal values, calculated assuming that DPPC/DDAB and cholesterol have the same 1/C_s_ in the ternary monolayer than in independent monolayers, according to Equation (7).
(7)$$\frac{1}{C_{s}{}^{\mathrm{ideal}}}=A_{\mathrm{molec}}^{\mathrm{ideal}}[x_{\mathrm{chol}}A_{{\mathrm{molec}(x}_{\mathrm{chol}}=1)}C_{s}{}_{{(x}_{\mathrm{chol}}=1)}+(1-x_{\mathrm{chol}})A_{{\mathrm{molec}(x}_{\mathrm{chol}}=0)}C_{s}{}_{{(x}_{\mathrm{chol}}=0)}]$$

The positive deviations from the ideal 1/*C_s_* can be observed in the x_chol_ ranges c.a. 0.2–0.5 and 0.7–0.9. In the lowest mole fraction range, this change involved 1/*C_s_* values higher than 100 mN/m at 30 mN/m, corresponding to the LC phase induced by cholesterol.

The influence of cholesterol in the stability of the monolayer can be analyzed in terms of the excess molecular area (ΔA_exc_), which is the difference between the averaged molecular area (A_molec_) and the ideal one ($A_{\mathrm{molec}}^{\mathrm{ideal}}$) calculated with Equation (6), ${\Delta A}_{\mathrm{exc}}{=A}_{\mathrm{molec}}-A_{\mathrm{molec}}^{\mathrm{ideal}}$ and the excess free energy of the monolayer (ΔG_exc_), defined in Equation (8).
(8)$${\Delta G}_{\mathrm{exc}}=\int_{0}^{\pi }{\Delta A}_{\mathrm{exc}}d\pi$$

These parameters are represented as a function of the mole fraction of cholesterol in Figure 4C,D.

### 3.3. Characterization of Liposomal Dispersions

The results obtained regarding d_H_, PDI, ζ and EE % of different liposomal formulations are summarized in Table 2. All liposomal dispersions, independently of their compositions, exhibited sizes in the nanometer scale, with a narrow size range between approximately 150 and 350 nm. As observed, the proportion of cholesterol influenced L size. All liposomal formulations with higher cholesterol levels showed lower sizes than L1 series with 3.3 mol% of cholesterol. AuNPs-containing liposomal formulations with and without D exhibited higher sizes than their respective counterparts without the AuNPs anchoring. PDI values were acceptable in all samples (PDI ≤ 0.3).

Higher ζ values were observed for L1 and L1-D in comparison to L2 and L2-D. The functionalization with AuNPs induced a reduction in ζ values of D-unloaded L. AuNPs anchoring and D loading greatly reduced the ζ values, achieving values of 1.6 and −1.9 mV for AuNPs-L1-D and AuNPs-L2-D, respectively. As observed in the sample with the highest cholesterol proportion, the functionalization with AuNPs as well as the D encapsulation induced a shift in the sign of ζ value from positive to negative.

Similar EE percentages were observed for both AuNPs-containing L (~78%), and these percentages were lower than the EE % observed in L1-D and L2-D. A higher amount of D loaded was determined in the liposomal formulation with the highest percentage of cholesterol (L2-D) compared to L1-D (94% and 84%, respectively).

The morphology of liposomal dispersions as well as the presence of AuNPs anchoring their surface were analyzed by FE-SEM and TEM (Figure 5). Lipid-based nanostructures exhibited approximately spherical shapes in the nanometer scale (from approximately 100 to 230 nm), and AuNPs decorating their surfaces were visualized in the corresponding samples. No noticeable differences were observed between the D-unloaded and loaded liposomal formulations. AuNPs as pure samples exhibited spherical shape and small sizes, and the AuNPs were not agglomerated.

### 3.4. In Vitro Doxorubicin Release

The in vitro release studies of D from liposomal formulations toward Hepes buffer solutions at pH 7.4 at 37 °C and 42 °C as receptor medium was performed. Figure 6 shows D release profiles from both liposomal dispersions with high and low cholesterol proportion and with and without AuNPs functionalization. The release profiles of a reference sample of pure D are also displayed.

From both L series, controlled D release towards Hepes buffer solutions at the two temperatures evaluated was observed, being more extended from L1 than from L2 series, whereas the release of D reference sample was faster, reaching almost 100% at 8 h of the assay. Similar release profiles for D from L1-D and AuNPs-L1-D were obtained, reaching 30–40% of drug released after 8 h of assay at 37 °C (Figure 6A). A slight increase in the percentage of D released was observed at 42 °C, wherein 50–60% was reached (Figure 6B). Lower percentages of D were released from L1 series compared to the L2 series. Different release profiles were observed in the samples with high proportion of cholesterol (L2 series). AuNPs-L2-D released a higher amount of D (~75%) compared to non-functionalized L (L2-D, ~55%) at 37 °C, after 8 h (Figure 6A), while at 42 °C the percentages reached were higher. Under this last condition, ~90% and 75% of D was released at that time (Figure 6B). As observed, the percentage of D released from L2-D was almost the same at both temperatures after 8 h. Most of the release profiles showed that at 24 h, the percentages of D reached were similar to those at 8 h, suggesting a steady-state condition.

Kinetic analyses of in vitro release data using Zero order, Higuchi and Peppas equations were performed to evaluate the main mechanism of D transport through the lipid-based nanostructures. Results of kinetic analyses are summarized in Table 3. Most drug release data plotted as Higuchi and Peppas model were found to be fairly linear for both of them, which was also supported by their regression coefficient values (R^2^ > 0.91 and R^2^ > 0.92, respectively). Furthermore, *n* values confirmed that the kinetics of D release from theL1 series, slightly lower or near to 0.5, indicated a Fickian transport with a preponderant release mechanism controlled by drug diffusion, in agreement with the good correlation coefficient obtained by application of the Higuchi model. For the L2 series, anomalous kinetic behaviors (n > 0.5) were determined, in which kinetic control was not only controlled by drug diffusion, but also involved other mechanisms (e.g., membrane permeability, disruption, etc.). In the case of the D reference sample, the release profiles were well fitted to Zero order (R^2^ > 0.99), suggesting a constant release rate.

### 3.5. In Vitro Cytotoxic Activity

The cytotoxic activity of the D-containing L was evaluated by using the resazurin proliferation assay. Because D is widely used to treat a variety of malignant tumors (e.g., breast, ovarian, lung and bladder cancer), we used the MDA-MB-231 breast cancer and the SK-OV-3 ovarian cancer cell lines for in vitro evaluation. To evaluate the influence of the release kinetic characteristics of D from L on cytotoxic activity, cells were treated for 2 h, followed by a recovery period of 70 h in drug-free medium; or cells were continuously exposed to drugs for 72 h.

As shown in Figure 7A,B, after 2 h treatment, the cytotoxicity of free D was higher than D released from L. The inhibitory concentration 50 (IC_50_) values (mean ± SEM, ng/mL) of free D in MDA-MB-231 and SK-OV-3 cells were 393.3 ± 51.0 and 323.0 ± 36.5, respectively. Lower cytotoxic effects were observed for D-loaded L compared to free D. According to the results shown in Figure 6, the L released approximately 20% of the drug during the first hour and 30–40% during the second hour at 37 °C. Indeed, AuNPs-L2-D showed slower D release than L2-D (Figure 6) and displayed less activity than L2-D in both cell lines (Figure 7A,B). Drug-unloaded L did not develop any cytotoxic activity.

Figure 7C,D shows the cytotoxic effects of free D and D-loaded L on cancer cell lines after 72-h treatment. Drug-unloaded L were non-toxic to the cells. Both D-loaded L (AuNPs-L2-D and L2-D) displayed cytotoxicity against cancer cell lines, suggesting that D was released from these L over time. L2-D showed greater inhibition of cell proliferation than AuNPs-L2-D, probably as a consequence of slower D release kinetic characteristics of AuNPs-L2-D at 37 °C. D-loaded L showed less cytotoxicity than free D. Concretely, the IC_50_ values (mean ± SEM, ng/mL) of D, L2-D and AuNPs-L2-D in MDA-MB-231 cells were 5.0 ± 1.4, 66.4 ± 9.4 and 118.7 ± 3.6, and in SK-OV-3 cells were 11.0 ± 0.1, 129.3 ± 4.4 and 196.0 ± 30.2, respectively.

## 4. Discussion

Even though pharmaceutical companies have focused their research on developing new formulations trying to reduce D-induced cardiotoxicity while keeping its efficacy, such as the D-containing liposomal formulations Caelyx^®^ and Myocet^®^ [46,47], these pharmaceutical products cause specific side events, such as hand–foot syndrome and stomatitis. Hence, there is still a need to develop new and optimized formulations to reduce systemic toxicity and increase or maintain the antitumor efficacy of D. Liposomal encapsulation has been demonstrated to be a good strategy (a) to reduce drug clearance by the hepatic, renal and immune system, increasing the drug availability, and (b) to specifically release the drug into tumor tissue, improving antitumor activity and avoiding damage to normal tissues [7,10,47]. Here, we report the design, development and evaluation of new AuNPs-functionalized L as nanocarriers for controlled D delivery.

The proportion of the lipid components used for preparing liposomal formulations is critical since it defines the stability of the obtained lipid-based nanosystems. In a previous study, we optimized the lipid vesicle composition of L-α-phosphatidylcholine-containing L by using cationic agents such as DDAB since as stabilizer and charge-inducer substance it allowed reinforcing the anchoring process of AuNPs onto the surface of L, and also provided stabilization to the vesicles [1]. Moreover, in another study, the vesicle-stabilizing effect of DDAB and cholesterol was demonstrated by the authors in L formed with the phospholipid 1,2-dimyristoyl-sn-glycero-3-phosphocholine (DMPC):DDAB in a 3:1 ratio and the optimum ratio of cholesterol, which was determined after analyzing the Langmuir isotherms of lipid monolayers containing different mole fractions of cholesterol [48]. Those results prompted us to evaluate the effects of cholesterol levels in the performance of AuNPs-anchored L composed of DPPC and DDAB as carriers for controlled delivery of D, a well-known anticancer agent. Thus, two series of liposomal formulations with low (L1) and high (L2) cholesterol levels were prepared. The low proportion of cholesterol was established based on previous results obtained by the authors [1,48]. The high proportion of cholesterol was evaluated in the current study for which monolayers at the air/water interphase were prepared and Langmuir isotherms of DPPC:DDAB:cholesterol ternary monolayers were measured to define the optimum mole fraction of cholesterol to prepare the L2 series.

It has been well established that values of 1/*C*_s_ in the range 12–100 mN/m indicate the existence of an LE state in the monolayer, while values in the range of 100–250 mN/m are characteristic of an LC state in the monolayer and values higher than 250 mN/m correspond to a solid state [49]. The values for 1/*C_s_* represented in Figure 3B indicate that the inclusion of DDAB in the formulation of the lipidic monolayer originates some disorder in the lipid molecules, changing from an LC state for pure DPPC to an LE state. This change must be a consequence of the electrostatic interactions between the positive charges of DDAB, which showed a lower dependence with the distance (1/r^2^) than the Lenard–Jones interactions at closed distances between the acyl chains of DPPC. Then, L formed with DPPC:DDAB, with the appropriate charge for anchoring of AuNPs may exhibit a fluidity too high for a drug carrier. The inclusion of cholesterol in the liposomal formulation increases the 1/*C_s_* values of the monolayer and, therefore, the LC of the monolayer, particularly at mole fractions of cholesterol, x_chol_, higher than 0.2.

In Figure 4A, it can be observed that in the x_chol_ range 0.2–0.6, the averaged area per molecule is lower than the ideal value, indicating some condensing effect of the cholesterol in the monolayer, clearer at lower surface pressures, caused by attractive intermolecular interactions. This condensing effect of cholesterol in the ternary monolayer has also been observed in binary DPPC/cholesterol monolayers in a much wider x_chol_ range [50].

The negative values of ΔA_exc_ in Figure 4C reveal the condensing effect of cholesterol, in a higher magnitude at low surface pressure. However, at 30 mN/m, which corresponded to the approximate equilibrium pressure of DPPC monolayers, the condensing effect was negligible. On the contrary, Figure 4D shows that the negative values for ΔG_exc_ exhibited a higher magnitude at 30 mN/m in the x_chol_ range from 0.3 to 0.5. This indicated that cholesterol in that mole fraction range stabilized the lipid film formed by DPPC/DDAB in the 3:1 molar ratio. Considering this stabilization and the LC-induced phase, the optimum mole fraction of cholesterol in the L composed by DPPC/DDAB/cholesterol was in the range of 0.3–0.5. Furthermore, to evaluate the effects of cholesterol levels in the performance of AuNPs-functionalized L, mole fractions of 3.35% and 40% were studied.

Besides the proportion of components to prepare the lipid-based nanocarriers, when thinking about drug loading through transmembrane pH-gradient method [41,42], the behavior of D solubility in different media and at different pH values is key to predict the state of the drug once encapsulated into the L. For that reason, different ammonium salts at two different pH values, simulating the acidic core (pH = 4) of vesicles and physiological conditions (pH = 7) were studied [43]. The transmembrane pH-gradient method was selected for encapsulating D since it allowed achieving high drug concentration within the L [51]. The use of ammonium salt is more effective than the sodium salt gradient since NH_4_^+^ acts as a reservoir, providing new free protons when D is protonated. In this method, uncharged D could easily penetrate and diffuse through the lipid bilayer to the internal aqueous compartment of L, facilitating its accumulation there, and it became protonated in the core. Once positively charged, D could no longer diffuse through the bilayer and it was trapped inside the vesicle [43] (Appendix A).

According to the results, in ammonium sulfate solutions, D solubility did not depend on the pH, being low at both pH values evaluated. In acidic pH and ammonium salts of phosphate and citrate, D remained highly soluble, while under physiological pH its solubility decreased; thus, large precipitates of D crystals were observed (Figure 1 and Figure 2). The obtained results agree with previous reports. After deprotonation, uncharged D was nearly insoluble in the salt solution, probably due to the cosmotropic properties of the counterion, phosphate or citrate, which contributed to the solubility of D [43]. In particular, for the interaction of D with sulfate anions, it has been reported that positively charged D precipitates with negatively charged sulfates to form a gel-like structure within the L. The cation (protonated D)–anion (sulfate) interaction in the solubility study showed the poor solubility of D in ammonium sulfate solution at a pH value close to the pK_a_ of the amino group of D (pK_a_ = 8.1) [52]. Moreover, crystal sizes (Figure 2) could predict the dissolution properties of drug precipitates. At pH 7, sulfate salt formed D precipitates with larger sizes than the other anions, which could anticipate a lower drug dissolution rate and thus a more controlled drug release. Taking into account these results, ammonium sulfate was the salt selected to perform the transmembrane pH-gradient, and high EE percentages were achieved (Table 2), demonstrating the suitability of this method in the development of the D-containing liposomal formulations [53]. The slightly lower values of EE % compared with previous reports can be explained considering the different bilayer composition [54] as well as the salt selected to generate the pH-gradient [43].

The lower size observed by DLS (Table 2) and FE-SEM (Figure 5) for the L2 series compared to the L1 series may be due to the high mole fraction of cholesterol, which could contribute to the bilayer rearrangement, favoring a reduction in the vesicle size [55]. The cholesterol proportion with respect to DPPC in membranes also influenced the EE % [27,28,29], which was more noticeable in the non-functionalized L. L2-D presented higher EE % than L1-D, probably due to the higher mole fraction of cholesterol. The high structuring property that this compound may confer to the bilayer could also increase the size of the aqueous compartment and, therefore, would facilitate an enhanced drug encapsulation within the core [56]. Moreover, AuNPs functionalization and D encapsulation into both non-functionalized and AuNPs-functionalized L increased their sizes compared to drug-unloaded and non-decorated L, respectively. Therefore, the higher the system complexity, the higher the d_H_ [54]. Anchoring process affected the EE %, and the reduced values observed in AuNPs-functionalized L compared to their non-anchored counterpart can be explained considering the destabilization that AuNPs can produce in the bilayer [57], which would generate leaking of a proportion of D encapsulated from the aqueous compartment.

Surface functionalization with AuNPs not only had an impact on vesicle sizes and EE % but also on ζ values, as expected. After anchoring, the ζ decreased compared to the respective non-functionalized L [58], especially for those with high mole fraction of cholesterol (L2), in which the ζ shifted from positive to negative.

FE-SEM images confirmed the almost spherical shapes in the nanometer scale of both L1 and L2 series, which was also confirmed by TEM analysis for L2 series, in agreement with previous works [19,59]. When comparing AuNPs-functionalized and non-functionalized L, some deformations in the surface membranes of L could be observed, which can be explained considering the different elastic interactions that interplay between the AuNPs and lipid membrane. It has been proposed that a cooperative absorption onto the outer surface of the liposomal membrane as well as wrapping events occurred with small AuNPs [59].

Both series of liposomal formulations allowed an efficient control of D release (Figure 6), demonstrating their promising behavior as carrier nanoplatforms for controlled drug release as well as their potential for hyperthermia therapy. The high proportion of cholesterol not only influenced the amounts of D released, which were higher for the L2 compared to the L1 series, but also the kinetic mechanisms. This can be explained considering that the high mole fraction of cholesterol increased the rigidity of liposomal membranes and induced a more ordered state of bilayers, influencing the drug release profiles [56,60]. Interestingly, release studies from AuNPs-functionalized L demonstrated a higher percentage of D released at higher temperatures (42 °C) compared to physiological temperatures (37 °C), confirming their thermal response. As has been reported, the fluidity of lipid membranes of L is enhanced at high temperatures, which increases the lipid bilayer permeability through larger channels on the surface of the vesicle, leading to a greater D diffusion compared to that observed at 37 °C [61]. Furthermore, release results also suggested that the mechanism behind the observed triggered release at high temperature could be due to the formation of transient pores in the bilayer or due to other forms of mechanical disruption of the lipid bilayer due to the AuNPs functionalization [62]. In fact, the behavior observed at 42 °C for the L2 series demonstrated the influence of the bilayer stability and the affinity of AuNPs to interact with the drug [63]. In addition, another study indicated that the high temperature favored the dissolution of D precipitates [64], which also could explain the increased percentages of drugs released at 42 °C. Kinetic data results were well fitted to Peppas model, and the observed behavior at high temperature agreed with previous results [65].

Considering the impact of high cholesterol mole fraction in the bilayer of L as well as their promising behavior observed in drug release studies, the L2 series was selected to perform cell studies. Cytotoxicity studies performed against human breast and ovarian cancer cells (MDA-MB-231 and SK-OV-3 cells, respectively) showed that the anticancer activity of AuNPs-L2-D was time-dependent. As shown in Figure 7, a short treatment was not enough to release the D from the L, and therefore, AuNPs-L2-D showed a very low cytotoxic effect compared to free D. This can be explained considering that the drug leaked very slowly due to loss of L gradient, which retained D, thus providing a slow diffusion of the drug to the cancer cells [61]. However, AuNPs-L2-D showed a marked inhibition of cell proliferation when cells were exposed for 72 h. These results, in agreement with release studies, suggest that D molecules entrapped within AuNPs-L2-D were released in a sustained manner over time and D still maintained its anticancer activity. Although the AuNPs-L2-D showed less cytotoxicity than L2-D, this difference was probably due to their slower D release rate at 37 °C. As is well-known, to grow, cells must be incubated at 37 °C, and AuNPs-L2-D were designed to release the anticancer agent at high temperatures (40–42 °C). Indeed, the in vitro release studies (Figure 6) showed that only ~65% of the D molecules entrapped within AuNPs-L2-D were released within 24 h at 37 °C compared to 80% in the case of L2-D. Although further studies are necessary to evaluate the efficacy and safety of AuNPs-L2-D, including photothermal therapy evaluation, the results obtained against breast and ovarian cancer cells not only confirmed the anticancer efficacy of developed liposomal formulations, emphasizing that D preserved its effect to combat cancer cells but also suggest a therapeutic potential of the AuNPs functionalization to increase the thermo-sensitive behavior of L. In this regard, this work contributes to the advancement of knowledge on hybrid nanocarriers based on AuNPs and L [15] and also to expand their potential for photothermal therapy [16,19] and as thermo-sensitive nanocarriers [8,9] for improving the treatment of cancer.

Summing up, our results showed that the developed D-loaded liposomal formulations would be a promising strategy for cancer therapy taking into account their nanometer size, capacity to control the drug release and functionalization with AuNPs, which could be exploited in hyperthermia treatments. The mole fraction of cholesterol in the bilayer positively impacted the drug loading, size and release behavior of the L2 series. The transmembrane pH-gradient method allowed achieving a high percentage of drug entrapped in the aqueous compartment of L. In vitro anticancer activity demonstrated that D-encapsulated L preserved its antiproliferative activity against cancer cells. Overall, the promising properties exhibited by the developed liposomal formulations could be further explored in other cell lines as well as in murine models in presence of external stimuli for inducing photothermal effects. Furthermore, studies to evaluate their efficacy and safety are also required to establish their advantages over the traditional treatments with D, mainly considering the cardiotoxicity and damage on red blood cells induced by D.

## Figures and Tables

**Figure 1 pharmaceutics-13-00973-f001:**
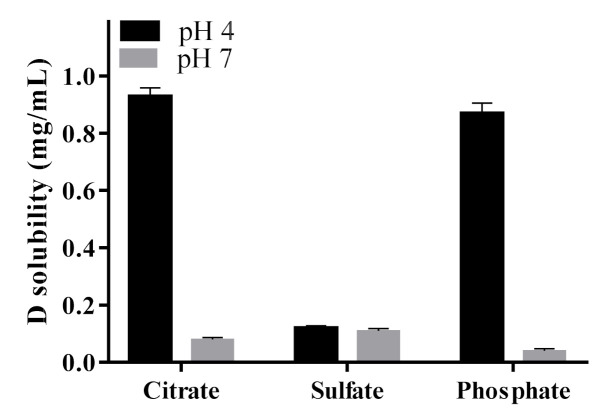
Solubility of doxorubicin (D) in different salt solutions as a function of pH.

**Figure 2 pharmaceutics-13-00973-f002:**
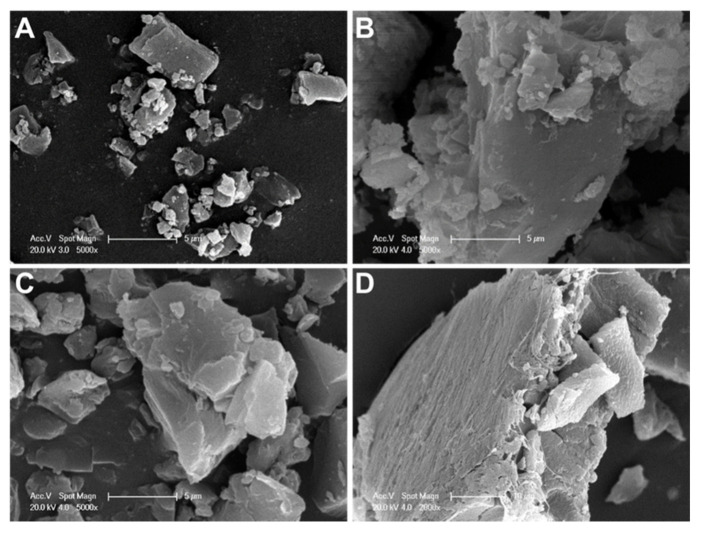
SEM images of different doxorubicin salts: (**A**) sulfate salt at pH 4, (**B**) sulfate salt at pH 7, (**C**) citrate salt at pH 7, and (**D**) phosphate salt at pH 7 (**A**–**C**, scale bar: 5 µm; **D**, scale bar: 10 µm).

**Figure 3 pharmaceutics-13-00973-f003:**
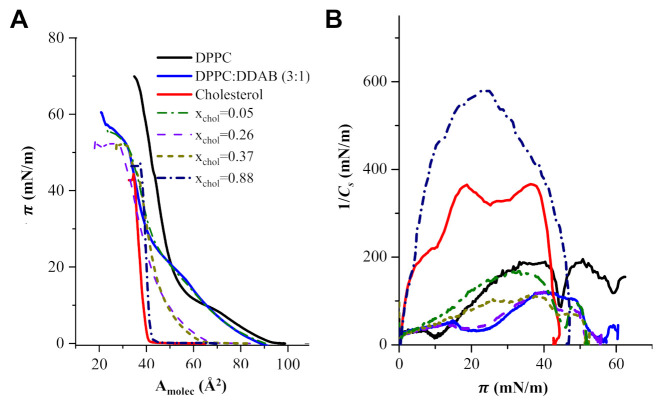
(**A**) Langmuir isotherms of monolayers of pure DPPC, DPPC:DDAB in a molar ratio 3:1, cholesterol and ternary monolayers of DPPC:DDAB 3:1 and cholesterol at some selected mole fractions of cholesterol (x_Chol_). (**B**) 1/*C_s_* vs. surface pressure plots for monolayers of pure DPPC, DPPC:DDAB in a molar ratio 3:1, cholesterol and ternary monolayers in panel (**A**).

**Figure 4 pharmaceutics-13-00973-f004:**
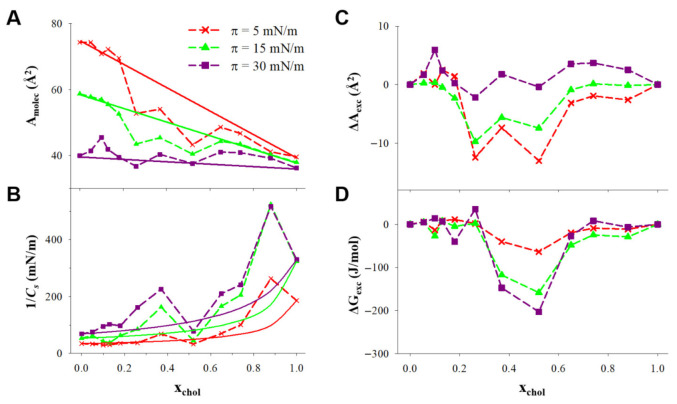
(**A**) Area per molecule and (**B**) compression modulus (1/C_s_) plotted vs. the cholesterol mole fraction at the constant pressures indicated in the figure for monolayers of DPPC:DDAB 3:1 and cholesterol. (**C**) Excess area per molecule (ΔA_exc_), and (**D**) excess free energy (ΔG_exc_) plotted vs. the cholesterol mole fraction at the constant pressures indicated in the figure for monolayers of DPPC:DDAB 3:1 and cholesterol.

**Figure 5 pharmaceutics-13-00973-f005:**
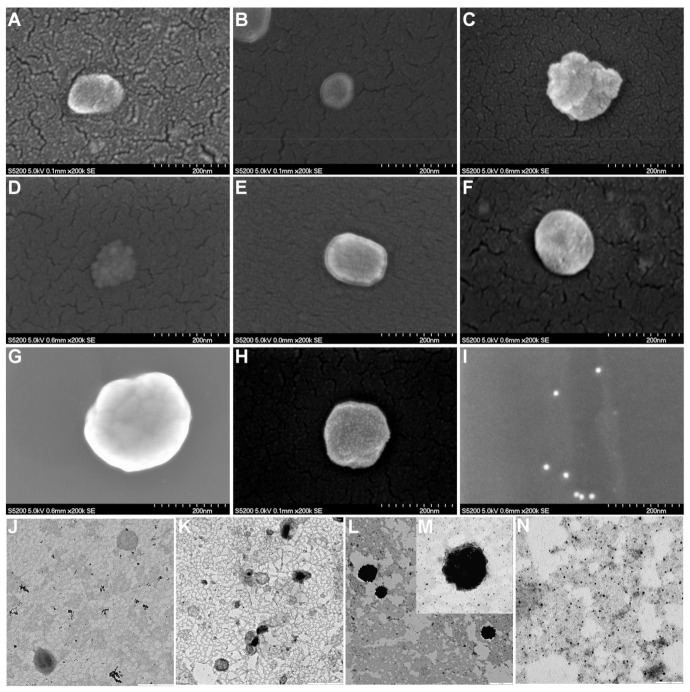
FE-SEM images of (**A**) L1, (**B**) L2, (**C**) AuNPs-L1, (**D**) AuNPs-L2, (**E**) L1-D, (**F**) L2-D, (**G**) AuNPs-L1-D, (**H**) AuNPs-L2-D, and (**I**) AuNPs (scale bar: 200 nm). TEM images of (**J**) L2 (scale bar: 0.5 µm), (**K**) L2-D (scale bar: 1 µm), (**L**,**M**) AuNPs-L2-D (scale bar: 0.5 and 0.1 µm, respectively) and (**N**) AuNPs (scale bar: 0.1 µm). L1 and L2: liposomes with different proportions of cholesterol, 3.35 mol% y 40 mol%, respectively; AuNPs: gold nanoparticles (15 nm), and D: doxorubicin.

**Figure 6 pharmaceutics-13-00973-f006:**
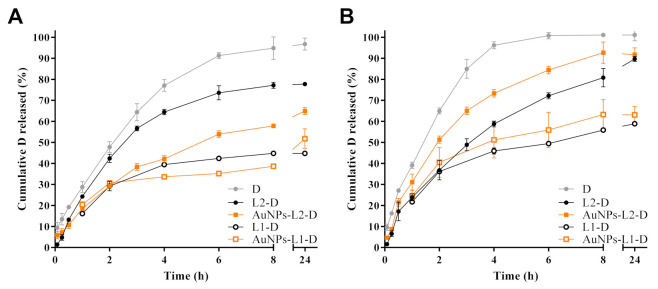
Effects of temperature, (**A**) 37 °C and (**B**) 42 °C, on in vitro release of doxorubicin (D) from both liposomes (L1 and L2, with different proportions of cholesterol, 3.35 mol% and 40 mol%, respectively) and gold nanoparticles (AuNPs)-functionalized liposomes toward Hepes buffer pH 7.4. In vitro D release profiles from the free drug are also shown.

**Figure 7 pharmaceutics-13-00973-f007:**
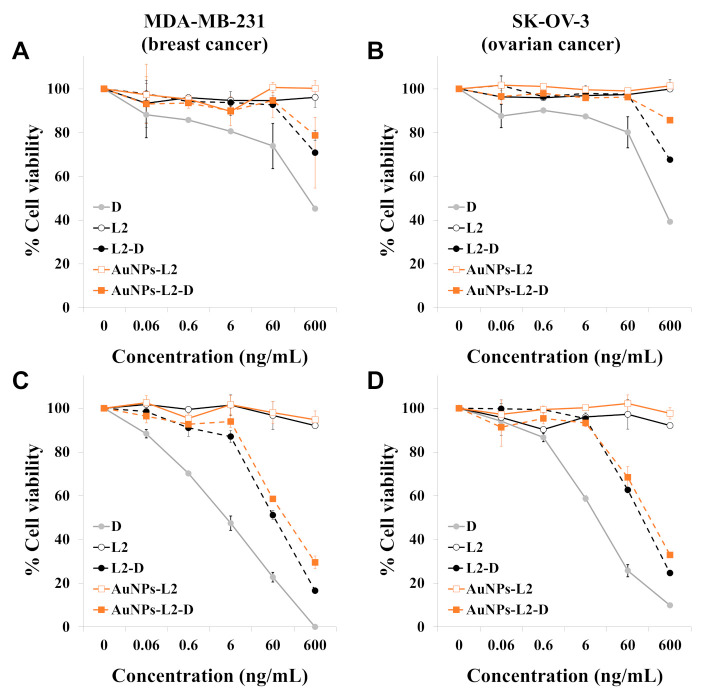
Evaluation of the cytotoxic activity of free doxorubicin (**D**), drug unloaded liposomes, non-functionalized and functionalized with gold nanoparticles (L2 and AuNPs-L2, respectively) and D-loaded liposomes (L2-D and AuNPs-L2-D, respectively). MDA-MB-231 breast cancer cells and SK-OV-3 ovarian cancer cells were treated for 2 h (**A**,**B**), followed by a recovery period of 70 h in drug-free medium, or cells were continuously exposed to compounds for 72 h (**C**,**D**, respectively). Finally, cell viability was determined by the resazurin assay. The concentrations of the unloaded L represented in the graphs are those used to deliver the concentrations of D shown on the x-axis of the graphs. Data are reported as means ± standard error of mean (SEM) and were obtained from two independent experiments. L2 corresponds to the liposome series with 40 mol% of cholesterol.

**Table 1 pharmaceutics-13-00973-t001:** Sample nomenclatures depending on the composition of the different preparations.

Sample	Chol (3.35)	Chol (40)	AuNPs	D
L1				
L2				
AuNPs-L1				
AuNPs-L2				
L1-D				
L2-D				
AuNPs-L1-D				
AuNPs-L2-D				

Chol (3.35) y Chol (40) correspond to the different proportions of cholesterol in the samples, 3.35 mol% y 40 mol%, respectively. AuNPs: gold nanoparticles (15 nm) and D: doxorubicin. The background color indicate the components at each sample.

**Table 2 pharmaceutics-13-00973-t002:** Encapsulation efficiency (EE %), hydrodynamic apparent diameter (d_H_), polydispersity index (PDI) and zeta potential (ζ) of both liposomal formulations with low (L1) and high (L2) cholesterol levels, loaded and unloaded with doxorubicin (D) and functionalized with AuNPs.

Sample	EE %	d_H_ (nm)	PDI	ζ (mV)
L1	–	245 ± 2	0.27 ± 0.01	18.8 ± 0.5
AuNPs-L1	–	320 ± 4	0.22 ± 0.02	15.7 ± 0.9
L1-D	84 ± 1	293 ± 4	0.24 ± 0.04	20.8 ± 0.7
AuNPs-L1-D	78 ± 2	347 ± 6	0.22 ± 0.03	1.6 ± 0.6
L2	–	149 ± 2	0.16 ± 0.02	14.7 ± 0.9
AuNPs-L2	–	293 ± 8	0.30 ± 0.02	7.4 ± 0.8
L2-D	94 ± 2	240 ± 4	0.25 ± 0.02	14 ± 1
AuNPs-L2-D	78 ± 3	304 ± 5	0.22 ± 0.03	–1.9 ± 0.9

**Table 3 pharmaceutics-13-00973-t003:** Kinetic data obtained from doxorubicin (D) release studies toward Hepes buffer pH 7.4 at two different temperatures, using Zero order, Higuchi and Korsmeyer-Peppas equations.

Temperature (°C)	Kinetic Models							
	Sample	Zero Order		Higuchi		Korsmeyer-Peppas		
		*k* _Z_	R^2^	*k* _H_	R^2^	*k* _P_	*n*	R^2^
37	D	18.8	0.998	38.5	0.978	30.8	0.65	0.996
	L1-D	3.7	0.823	15.1	0.910	20.4	0.40	0.922
	AuNPs-L1-D	2.2	0.787	8.7	0.863	23.1	0.25	0.960
	L2-D	16.5	0.979	39.1	0.994	23.6	0.76	0.983
	AuNPs-L2-D	6.9	0.930	22.3	0.991	19.4	0.55	0.998
42	D	28.3	0.990	39.3	0.988	41.0	0.65	0.996
	L1-D	4.4	0.881	17.4	0.944	25.4	0.39	0.956
	AuNPs-L1-D	5.0	0.894	19.8	0.954	28.4	0.39	0.963
	L2-D	13.2	0.976	33.2	0.996	23.1	0.68	0.992
	AuNPs-L2-D	20.6	0.971	43.1	0.994	31.0	0.69	0.993

*k_Z_*, *k_H_* and *k_P_* expressed as % h^−1^, % h^−0.5^ and % *k*^−n^, respectively. Experimental data correspond to 5–60% of D released. L1 and L2: liposomes with different proportions of cholesterol, 3.35 mol% y 40 mol%, respectively; AuNPs: gold nanoparticles.

## Data Availability

Not applicable.

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
