# Peer review of "Cholesterol Levels Affect the Performance of AuNPs-Decorated Thermo-Sensitive Liposomes as Nanocarriers for Controlled Doxorubicin Delivery"

_pharmaceutics, 2021, doi:10.3390/pharmaceutics13070973_

Round 1

Reviewer 1 Report

The effect of cholesterol levels on liposomes formation and drug release is clear. However, the biological results are weak. No enough biological results support the proposed novelty of this investigation. The following suggestions may help improve the manuscript.

  1. The novelty of this research is thermal-triggered amplification of drug release for thermo-sensitive liposomes. Indeed, recently,  drug release is identified as key contributor for efficacy of nanomedicine (J. Am. Chem. Soc. 2021, 143, 2, 538–559).  The authors should emphasize this point. The novelty based on the effect of cholesterol levels is not strong.
  2. The authors should study the photo-thermal effect of Au-liposomes. Then also the amplification of  drug release (doxorubicin) and cytotoxicity from photo-thermal effect should be proved. Some investigations already proposed this problem and provided the solutions (Cancer Res. 2012, 72 (21), 5566−75; J. Controlled Release 2009, 137 (1), 63−8; J. Controlled Release 2016, 225, 64−74; Nano Lett. 2017, 17 (11), 6983−90.). The authors refer them and provide some experiments on this point. This is key point of this investigation.
  3. Cryo-TEM or TEM should be provided to demonstrate the real vesicular structure of Au-liposomes. Just SEM can not give the clear information.
  4. UV-vis spectra should be provided for Au and Au-liposomes. 

Author Response

POINT-BY-POINT REPLIES TO Reviewer #1 COMMENTS:

General Comments. The effect of cholesterol levels on liposomes formation and drug release is clear. However, the biological results are weak. No enough biological results support the proposed novelty of this investigation. The following suggestions may help improve the manuscript.

Answer to general comment. We understand the referee suggestion and appreciate her/him consideration related to incorporate more biological studies. However, considering the goal of the present work we consider that all the results and discussion presented contribute to the achievement of the aim proposed. We comprehensively evaluated the effect of cholesterol levels on the interfacial and morphological properties of thermo-sensitive liposomes as well as in their encapsulation efficiency and capacity to control the drug release. The selected mole fractions of cholesterol were rationally established based in thermodynamical studies (Langmuir isotherms). Once evaluated the physicochemical and pharmaceutical properties of both series of liposomes (with low and high levels of cholesterol) we advanced and evaluated their biological performance against two different cancer cell lines (human breast and ovarian cancer cells, MDA-MB-231 and SK-OV-3, respectively) at two different times (2 and 72 h), bearing in mind the relevance of this type of studies when using an anticancer drug as encapsulating agent. From these studies we confirmed that the drug was active against cancer cells, preserving its antiproliferative activity when carried into liposomes. We know that further studies are still required to biologically characterized the liposomal formulations, including new cells lines and photothermal therapy evaluation as well as in vivo studies. We are planning to perform different experiments in the near future to fully evaluate the bio-performance of the developed liposomes on other cell cultures as well as in animal models (xenografts or orthotopic) in presence and absence of light to define their potential use for photothermal therapy. Once obtained, those results will be organized in a future manuscript, in which the aim will be focused mainly in the biological performance of liposomal formulations.  

Comment 1. The novelty of this research is thermal-triggered amplification of drug release for thermo-sensitive liposomes. Indeed, recently, drug release is identified as key contributor for efficacy of nanomedicine (J. Am. Chem. Soc. 2021, 143, 2, 538–559).  The authors should emphasize this point. The novelty based on the effect of cholesterol levels is not strong.

Answer 1. Following the referee suggestion, the suggested Perspective article was analyzed and some relevant sentences (pg. 2) have been included as part of the introduction section in the revised version of the manuscript. Also, this published article was also considered for the discussion (pg. 19). Consequently, the respective reference was appropriately included in the manuscript and added to the list of references.

Comment 2. The authors should study the photo-thermal effect of Au-liposomes. Then also the amplification of drug release (doxorubicin) and cytotoxicity from photo-thermal effect should be proved. Some investigations already proposed this problem and provided the solutions (Cancer Res. 2012, 72 (21), 5566−75; J. Controlled Release 2009, 137 (1), 63−8; J. Controlled Release 2016, 225, 64−74; Nano Lett. 2017, 17 (11), 6983−90.). The authors refer them and provide some experiments on this point. This is key point of this investigation.

Answer 2. Considering this reviewer’s comment, we have read the suggested published articles. Despite all of them are more than interesting and many of them are related to (stimuli-responsive) nanocarriers for cancer therapy, we consider that they are not suitable for our manuscript and, particularly, they are not directly related to photothermal therapy and thermo-responsive liposomes. One of them (Cancer Res. 2012, 72 (21), 5566−75) even when it involved liposomal formulations loaded with doxorubicin in which the drug release was triggered by local heat, it mainly reported on in vivo studies, which has not been included in our manuscript.

Following the referee suggestion, we have performed a preliminary release study to evaluate the amplification of doxorubicin release under NIR irradiation, taking into consideration the potential of liposomes for photothermal therapy. In this study, we comparatively evaluated the percentage of drug released at three different times in presence and absence of NIR irradiation. Experiments were performed through a dialysis method (dialysis bag containing 1 mL of sample). Two different samples were analyzed, namely AuNPs-L2-D y L2-D, to evaluate the effect of surface anchoring with AuNPs. 50 mL of Hepes pH 7.4 were used as release medium. The whole setup was placed in an automated shaker and gently stirred at 80 rpm and 37 °C. The samples were subjected to two different conditions. On the one hand they were totally protected from light along the assay and, on the other hand, they were irradiated at determined time points (the second, the fourth and the eighth hour) for 5 min using 1100 nm lamp (IR 1, OSRAM Siccatherm, 250 W, 230 V). Samples of 0.5 mL of receptor medium were withdrawn at predetermined time intervals and replaced with equal quantities of fresh medium. The concentration of released doxorubicin was assayed in a fluorescence spectrophotometer (Hitachi® F-2500) at 480 and 580 nm of excitation and emission wavelengths, respectively. Furthermore, the temperature of the release media was measured after the 5 min of irradiation.

The results indicated that at the three times evaluated L2-D did not have noticeable differences in the percentage of drug release in absence and presence of irradiation. As expected, AuNPs-functionalized liposomal dispersions (AuNPs-L2-D) exhibited a 1.3-fold and 1.2-fold higher release when irradiated compared to non-irradiated samples at 2 and 4 h, respectively. At 8 h the percentage of drug released was similar under NIR irradiation compared to the respective sample in darkness. In addition, the mean temperature reached in the release medium after NIR irradiation was 43.5 °C. These results, although preliminary, demonstrated the effect of the anchoring with AuNPs and their sensitive at NIR irradiation, emphasizing their light-inducing heating behavior, and also highlight the potential of developed liposomal formulation to be used for photothermal therapy.

Even though we performed this preliminary study and the results indicated the photothermal effect on amplification of drug release, further studies are still required to better evaluate the photothermal performance of developed liposomes. For these experiments, several parameters need to be studied to clearly define the intensity of the external source, the time and pulse of application, the distance between the samples and the light, the reproducibility of experiments, the ratio between AuNPs and liposomes, etc. These adjustments are required to find the best experimental conditions. For this reason, we decided do not include these preliminary results in our manuscript. We prefer advance on this research in the near future, as previously mentioned, and organize the new results in a future manuscript since we considered that the aim of the present manuscript has been achieved with the presented results.

Regarding studies of photo-thermal effect on cells, as we mentioned above, we are planning to perform different experiments in the near future to fully evaluate the biological properties of the developed liposomes not only against cancer cells but also in animal models in presence and absence of light to better define their potential use for photothermal therapy. Once obtained, those results will be organized in a future manuscript, in which the aim will be focused mainly in the bio-performance of thermo-sensitive liposomal formulations.  

Comment 3. Cryo-TEM or TEM should be provided to demonstrate the real vesicular structure of Au-liposomes. Just SEM can not give the clear information.

Answer 3. Following the referee suggestion, the sample set: L2, L2-D, AuNPs-L2-D and AuNPs were analyzed by TEM.

Accordingly, the methodology used (pg. 6, section 2.8.2. Morphological analysis), the obtained results (section 3.3. Characterization of liposomal dispersions, pg. 12 and Figure 5) and discussion (pg. 18) have been included in the revised version of the manuscript.

Comment 4. UV-vis spectra should be provided for Au and Au-liposomes.

Answer 4. Following the referee suggestion, the curves of UV-Vis spectra were measured.

Accordingly, the methodology used has been included in the revised version of the manuscript and the obtained results has been incorporated in the supplementary material.

The spectra showed the presence of the surface plasmon of disperse AuNPs, in agreement with the photothermal activity related to answer 2. However, partial aggregation could be inferred from the appearance of plasmon bands at higher wavelengths. In a future study, we are considering to balance the ratio AuNPs/liposomes in order to improve the optical properties of the liposomal formulation.

Reviewer 2 Report

In this work, effect of cholesterol concentration on activity of AuNPs-decorated thermos-sensitive liposomes for the controlled doxorubicin (as cancer treatment pharmaco-therapeutics) delivery, was studied.

REMARKS

INTRODUCTION (note)

I miss specially to this work related to DOX and breast cancer cells investigation. Please, follow the upgrade.

INTRODUCTION (upgrade, last paragraph)

In this study, doxorubicin (D), an anthracycline antibiotic originally isolated from Streptomyces peucetius var. caesius, was selected since it is one of the most commonly employed anticancer drugs. This drug is widely used in the treatment of solid and hematologic neoplasms, such as breast, ovarian, bladder cancer and lymphoma [https://doi.org/10.1515/biol-2019-0070].

MATERIALS AND METHODS (note)

Please, draw a procedure´s particular steps and provide it as a picture, to show the reader a quick view of what was done in this study in order to get a rapid comprehending.

CONCLUSION (note)

Please provide some options toward the future as for this kind of investigation.

Author Response

POINT-BY-POINT REPLIES TO Reviewer #2 COMMENTS:

General Comment. In this work, effect of cholesterol concentration on activity of AuNPs-decorated thermos-sensitive liposomes for the controlled doxorubicin (as cancer treatment pharmaco-therapeutics) delivery, was studied.  

Comment 1. I miss specially to this work related to DOX and breast cancer cells investigation. Please, follow the upgrade. INTRODUCTION (upgrade, last paragraph): In this study, doxorubicin (D), an anthracycline antibiotic originally isolated from Streptomyces peucetius var. caesius, was selected since it is one of the most commonly employed anticancer drugs. This drug is widely used in the treatment of solid and hematologic neoplasms, such as breast, ovarian, bladder cancer and lymphoma [https://doi.org/10.1515/biol-2019-0070].

Answer 1. Considering this reviewer’s comment, we have read the suggested published article. Despite it is more than interesting, we consider that it is not suitable for our manuscript since the main finding reported was obtained in MCF-7 cells while the breast cancer cells used in our study were MDA-MB-231. Even though we included two references almost at the end of the mentioned paragraph, to emphasize the information provided and support it with an appropriate reference, we included the reference from the National Cancer Institute-NIH at the indicated part of the introduction section (pg. 3).

Comment 2.  Please, draw a procedure´s particular steps and provide it as a picture, to show the reader a quick view of what was done in this study in order to get a rapid comprehending

Answer 2. Following the referee suggestion, a schematic depiction of the different steps involved in the preparation of the liposomal formulation has been included as supplementary material (see Figure S1) and it has been appropriately indicated in sections 2.5. Liposome preparation, 2.6. Doxorubicin encapsulation and 2.7. Functionalization with AuNPs (pg. 5-6)

Comment 3.  Please provide some options toward the future as for this kind of investigation.

Answer 3. Considering this reviewer’s comment, we expanded a little bit more the perspective of this work in the revised version of our manuscript (pg. 20).  

As mentioned in the manuscript, our work contributes to the advancement of knowledge on hybrid nanocarriers based on AuNPs and liposomes, and also to expand their potential for photothermal therapy and as thermo-sensitive liposomes for improving the treatment of cancer (pg. 19). In this regard, studies related to photothermal therapy are still required to better define the drug release performance from liposomal formulations under light-inducing heating. Further studies on other cancer cell cultures as well as in animal models in presence of external stimuli (temperature and light) would provide very useful data regarding the biological properties of developed liposomes. Their efficacy and safety need to be comprehensively evaluated to establish their advantages over the traditional treatments with D, mainly considering the cardiotoxicity and damage on red blood cells induced by D (pg. 20).

Round 2

Reviewer 1 Report

It can be published.

Author Response

POINT-BY-POINT REPLIES TO Reviewer #1 COMMENTS:

Comment. English language and style are fine/minor spell check required. It can be published

Answer. We are glad to know that our effort to respond the different comments raised in Round 1 were successfully addressed. Taking into consideration the comment related to English language and style, we carefully and comprehensively check once again our manuscript and correct the fine/minor spell. Also, other few minor changes were introduced in the manuscript to improve its clarity. 

Reviewer 2 Report

Regarding author´s response to comment 1, such text upgrade with citation refers in this case to general usage of DOX to treat different cancers. In this course, please follow the comment 1.

Comment 1. I miss specially to this work related to DOX and breast cancer cells investigation. Please, follow the upgrade. INTRODUCTION (upgrade, last paragraph): In this study, doxorubicin (D), an anthracycline antibiotic originally isolated from Streptomyces peucetius var. caesius, was selected since it is one of the most commonly employed anticancer drugs. This drug is widely used in the treatment of solid and hematologic neoplasms, such as breast, ovarian, bladder cancer and lymphoma [https://doi.org/10.1515/biol-2019-0070].

Author Response

POINT-BY-POINT REPLIES TO Reviewer #2 COMMENTS:

Comment. Regarding author´s response to comment 1, such text upgrade with citation refers in this case to general usage of DOX to treat different cancers. In this course, please follow the comment 1. Comment 1. I miss specially to this work related to DOX and breast cancer cells investigation. Please, follow the upgrade. INTRODUCTION (upgrade, last paragraph): In this study, doxorubicin (D), an anthracycline antibiotic originally isolated from Streptomyces peucetius var. caesius, was selected since it is one of the most commonly employed anticancer drugs. This drug is widely used in the treatment of solid and hematologic neoplasms, such as breast, ovarian, bladder cancer and lymphoma [https://doi.org/10.1515/biol-2019-0070].

Answer. Considering this reviewer’s comment, we have read once again the suggested published article and we have introduced the reference at the indicated part of the introduction section (pg. 3). Therefore, the paragraph was changed as follow:

In this study, doxorubicin (D), an anthracycline antibiotic originally isolated from Streptomyces peucetius var. caesius, was selected since it is one of the most commonly employed anticancer drugs. This drug is widely used in the treatment of solid and hematologic neoplasms, such as breast, ovarian, bladder cancer and lymphoma [33].

Ref. 33: Bober, P.; Alexovič, M.; Tomková, Z.; Kilík, R.; Sabo, J. RHOA and mDia1 promotes apoptosis of breast cancer cells via a high dose of doxorubicin treatment. Open Life Sciences 2019, 14, 619-627.

Round 3

Reviewer 2 Report

I consider publication of the manuscript at current state.